# A Prospective Observational Study of a 2-Week Integrative Inpatient Therapy on Patients with Fibromyalgia Syndrome

**DOI:** 10.3390/biomedicines13092144

**Published:** 2025-09-02

**Authors:** Sandra Utz, Christine Uecker, Stefanie Kropač, Jost Langhorst

**Affiliations:** 1Department for Integrative Medicine, Medical Faculty, University of Duisburg-Essen, 45117 Essen, Germany; sandra.utz@sozialstiftung-bamberg.de (S.U.); christine.uecker@sozialstiftung-bamberg.de (C.U.);; 2Department of Internal and Integrative Medicine, Sozialstiftung Bamberg, 96049 Bamberg, Germany

**Keywords:** fibromyalgia, integrative medicine, multimodal, medical fasting, pain, fatigue, quality of life

## Abstract

**Background**: The fibromyalgia syndrome (FMS) is classified as a functional somatic syndrome and is characterized primarily by chronic pain in multiple body regions and physical and/or mental fatigue. The German S3-guideline recommends a multimodal therapy for severe forms. Since research on non-pharmacologic complementary, naturopathic, and integrative therapy approaches shows positive and promising effects, integrative methods are firmly anchored in the S3 guideline. **Objective/Methods**: Aim of the present study was to investigate whether a multimodal integrative treatment program can be effective in reducing the primary symptoms of FMS (pain and fatigue) and improving psychological aspects such as quality of life (QoL), anxiety, depression, and perceived stress. Another aim of the study is to explore whether potential effects appear only in the short term (immediately after discharge) or persist long term (six months after discharge). The treatment concept is based on mind–body medicine and elements of classical European naturopathy (including fasting interventions) and focusses on stress reduction and lifestyle modification. **Results**: Of N = 134 originally included longstanding fibromyalgia patients (mean time since diagnosis 9.2 ± 8.5 years), 101 data sets could be analyzed. Results show a significant improvement in both short-term and long-term pain and fatigue intensity (about 12% improvement). Long-term reductions in pain intensity appear to be supported by medical fasting interventions. Regarding psychological aspects and quality of life, there are long-lasting reductions regarding anxiety, depression, perceived stress, and helplessness and a long-lasting increase in self-efficacy, quality of life and current working ability. **Conclusions**: The two-weeks inpatient stay therefore leads to significant improvements in all mentioned aspects. Therefore, the concept may be a promising component for integration into medical guidelines and thus in the care of FMS patients. Future research including randomized controlled trials is necessary to further evaluate the program.

## 1. Introduction

Fibromyalgia syndrome (FMS) is the third most common rheumatic disease, with a prevalence in Europe of approximately 2.3% to 2.5% [1,2]. In Germany, estimates range from 2.1% [3] to 2.7% [2]. Although FMS is incurable, it does not reduce life expectancy [4]. Its prognosis, however, depends on various factors, including demographic, behavioral, and psychological influences. Factors such as female gender, low socioeconomic status, unemployment [5,6], depression, stress, and obesity [7] are associated with poorer disease outcomes. Conversely, strong social support [8], physical activity [9], and early diagnosis, which facilitate better disease/self-management and reduced medication use [4,10,11], are associated with more favorable outcomes. Psychological elements, such as trauma, stress, and depression, play a critical role in FMS [12]. Depressive and sleep disturbances are bidirectionally linked to the condition [7,13]: Depression and/or psychosocial stressors (e.g., work, family) increase the risk of developing FMS, while chronic pain of FMS can exacerbate depression [14,15].

FMS is characterized by a diverse clinical picture and symptom complex. These include persistent pain across various regions of the body, fatigue, sleep disturbances, and frequent comorbid conditions like depression and anxiety (high comorbidity; [15,16,17,18]). Cognitive deficits, lack of motivation, and a need for extended recovery times after physical, mental, or emotional exertion are also common [19]. The complexity of diagnosing FMS has its origin in its unknown etiology and varied symptomatology. Diagnosis relies on a thorough assessment of medical history, physical examination, and the exclusion of other conditions. Tools like pain sketches and the Fibromyalgia Symptom Questionnaire can help evaluate the severity of the condition [20,21]. Comprehensive evaluations are crucial, particularly in addressing comorbidities like depression and anxiety [11]. The persistent pain and associated symptoms greatly impair the QoL for individuals with FMS. Daily work and leisure activities become burdensome, limiting social and societal participation. Work disability or absenteeism due to FMS is common, with an average of three to four weeks of work lost annually because of pain [22,23,24,25]. Consequently, FMS is not just a personal burden for individuals but also a societal issue due to the strain it places on healthcare systems and the economy.

The multifaceted nature of FMS suggests that its treatment is not straightforward. A variety of therapeutic approaches are available, and the choice of therapy often depends on the familiarity of the treating physician with current guidelines and the resources available [26]. Current guidelines emphasize the need for psychoeducation early in treatment, as well as fostering self-confidence and self-efficacy (S3 guideline; [27]). Self-efficacy, in particular, is a key factor in achieving successful treatment outcomes for pain and depression [28] and is a fundamental requirement for sustained therapy outcomes [10,29,30]. For patients with milder forms of FMS, physical and psychosocial activation (e.g., engaging in hobbies) is recommended. In more severe cases, a multimodal therapy concept is advised. This typically includes a combination of physical activity, psychotherapeutic approaches, and initial medication therapy (not necessarily required; [31]). The overall consensus is that endurance training, along with mild to moderate strength training (e.g., walking), is beneficial.

Given the high prevalence of physical and mental impairments among FMS patients, it is essential to treat these conditions according to current guidelines. Although no specific medication is approved for FMS in Germany, a multidimensional approach is seen as more effective than pharmacological treatments alone [32,33]. Although evidence supporting complementary medicine approaches remains limited [34], certain complementary therapies are recommended by the current guideline: the use of meditative and mindfulness movement therapies (i.e., tai chi, yoga, Qigong); weight loss or calorie-reduced diet in cases of obesity; time-limited use of electroacupuncture; mindfulness-based stress reduction (MBSR) used within a multimodal program. Studies show that treatment effectiveness varies across FMS patients, suggesting the need to tailor therapies to the specific characteristics of subgroups within the patient population (S3 guideline; [27,35]). It is therefore difficult to interpret the results of randomized controlled trials with random samples and measurements of the average effectiveness of the treatment in FMS patients. Such results could even be misleading [11].

Meditative and mindfulness-based movement therapies have been shown to reduce pain and can be practiced independently [36]. Tai chi, in particular, has demonstrated promising results in improving FMS symptoms, with longer practice periods leading to even greater benefits [37]. Similarly, an eight-week structured yoga program has been found to improve pain, fatigue, psychological well-being, and overall functionality in women with FMS [38]. Regular Qigong practice has shown improvements in pain, sleep quality, and mental and physical performance [39]. Incorporating mindfulness-based therapies from a subset of complementary medicine modalities, known as mind–body medicine (e.g., mindfulness exercises), as well as regulatory therapeutic interventions (e.g., breathing techniques), and progressive muscle relaxation can also help reduce anxiety, depression, and psychological stress and increase QoL [40]. The classic eight-week Mindfulness-Based Stress Reduction Program (MBSR) appears not to be superior to cognitive-behavioral therapy in terms of pain and depression reduction in chronic pain patients, but it could be a good alternative [41]. Certain dietary changes, such as adopting anti-inflammatory diets (Mediterranean, vegetarian, vegan, ketogenic), have been linked to reduced pain in chronic conditions [42], although evidence in FMS is still developing. Fasting under medical supervision (3–12 days, <600 kcal/d) has shown promising results in improving subjective disease impact, pain, and QoL, with long-lasting effects observed in some cases (up to nine months) [43,44,45].

The complexity of FMS requires different treatments that can be prioritized depending on the individual patient’s symptoms. For example, patients who are more depressed, inactive, and overweight may benefit from physical activation therapies, while those who experience burning pain and sensory disturbances may require treatments that address restlessness and exhaustion. Integrative therapies offer a personalized approach that takes into account the diverse symptomatology of FMS [10] and many patients therefore seek complementary and integrative medicine (CIM) therapies [46,47,48]. Integrative medicine involves in addition to conventional medicine naturopathic therapy elements (originally hydrotherapy, thermotherapy, phytotherapy, exercise therapy, nutritional therapy, and lifestyle counseling), as well as complementary treatments like acupuncture [49]. These approaches emphasize self-care, stress resilience, and self-management [10]. For long-term therapy, patients are encouraged to adopt self-management practices like endurance training, strength exercises, and heat therapy (S3 guideline; [27]).

Waterfiltered infrared-A whole body hyperthermia (WBH) is a special form of physical heat therapy. The mild form of WBH causes an increase in body core-temperature (<38.5 °C) and leads to an increased tissue perfusion, an acceleration of metabolic processes, has effects on immunological processes, and probably even on pain receptors [50]. A randomized controlled study (application of six WBH-treatments within three weeks compared to a control group with sham intervention) in FMS patients showed a significant reduction in pain intensity in the treatment vs. control group. The pain reduction was seen at the end of treatment (week 4) and at follow-up (30 weeks) [51].

Multimodal integrative approaches show promise in addressing the central symptoms of FMS, such as pain, fatigue, anxiety, and depression, which significantly affect quality of life. While further research is needed to solidify the evidence base for many of these therapies, initial findings suggest that integrative treatment can be particularly beneficial for chronic rheumatologic and pain conditions like FMS [35,52,53,54,55,56,57,58]. Previous studies have already shown positive effects of individual complementary or integrative medical therapies for FMS (e.g., tai chi [37], yoga [38], meditation [59], whole-body hyperthermia [50,51], nutritional therapy [60], phytotherapeutics [61]), but the application of a comprehensive integrative medical inpatient concept in the treatment of FMS has not yet been scientifically investigated.

## 2. Materials and Methods

### 2.1. Study Design

As part of ongoing research, the present prospective monocentric uncontrolled observational study is investigating the impact of coordinated complementary therapies on FMS patients during their two-weeks inpatient stay (the costs of the inpatient stay are covered by the patients‘ health insurance). At the clinic for Internal and Integrative Medicine in Bamberg, non-pharmacological therapies for FMS are tailored to individual patient needs [10,35,62,63,64]. The multimodal integrative therapy concept is based on the five pillars of classical European naturopathy (hydrotherapy, exercise therapy, nutrition, phytotherapy, and mind–body-medicine; [53,54,55,56,57,58,65]). In addition, methods of extended naturopathy like acupuncture are applied [56]. The therapies used in Bamberg include (among others) meditative movement and physical therapy to actively engage and passively activate physically inactive (potentially overweight) and/or depressed patients while alleviating pain. Hydrotherapy and thermotherapy (including waterfiltered WBH; if indicated, patients received one or two) can have both activating and relaxing effects, thereby reducing stress. Mind–body medicine approaches aim at teaching a health-promoting approach to the chronic illness. It focuses on teaching self-care strategies by specifically addressing interactions among emotional, mental, social, and behavioral factors. One focus is on relaxation techniques to help with stress and anxiety. Nutrition therapy (offering a plant-based, Mediterranean whole-food diet, or medical fasting), lectures, workshops, and cooking classes are provided to teach healthy eating. Fasting under medical supervision (1 preparatory day <600 kcal/d, 5 days fasting period <300 kcal/d, 3 refeeding days) has already shown improvements, among others, in pain [43,44,45]. Each patient was assessed individually to determine whether medical fasting was indicated or if there were some contraindications. For those with sleep problems and as a result suffer from severe physical and mental exhaustion, self-help strategies such as dry brushing according to Kneipp or rubs with rosemary oil, mallow oil, or Aconitum pain oil are offered. Phytotherapeutics can also be used to improve sleep disturbances using teas, sleep-promoting wraps (lavender heart compress), and internally (valerian, lavender oil capsules, melissa) alongside accompanying acupuncture (either auricular or body). This study aims to evaluate the short- and long-term effects of a multimodal integrative therapeutic approach on pain and other core FMS symptoms, as well as its impact on patients’ quality of life, anxiety, depression. Measurements are taken at baseline, discharge, and follow-up intervals. Furthermore, patients’ experiences with the implementation of the newly learned strategies into everyday life are assessed. The comprehensive research project “InteChron-Integrative Therapy for Chronic Diseases” received approval from the Ethics Committee of the Bavarian State Medical Association in Munich and is registered on clinicaltrials.gov (Identifier: NCT04927403). It was conducted in accordance with the current principles of the Declaration of Helsinki.

### 2.2. Participants

Recruitment of FMS patients was conducted during registration for admission to the inpatient stay by informing patients about the study and asking for their willingness to participate. Inclusion criteria were an age of at least 18 years, a proven FMS diagnosis, and a signed informed consent. Anamnesis upon admission to the ward was essential, so that if not previously performed by a physician, the diagnosis of FMS could be made based on the current diagnostic criteria. For those generally open to participation, a detailed personal instruction took place, and patients received written information about the study, including data protection notices. It was explained that participation or non-participation in the study would not affect their treatment, that they could withdraw from the study at any time without providing reasons for it, that there would be no additional costs associated with participating in this clinical trial, and that they would receive no financial compensation for participation. Patients interested in participating, had to give their written informed consent before they were included in the study.

### 2.3. Material

Patients completed all reported questionnaires at T1, T2, and T3, except for Satisfaction with treatment (only T2) and Implementation in daily life, which was only of interest at T3.

#### 2.3.1. Symptom-Specific Questionnaires

*Von Korff Grading of Severity questionnaire*. To assess pain intensity, the Von Korff Grading of Severity questionnaire [66] was used. Overall pain intensity was calculated using the first three items (1a to 1c). Item 1a captures current pain intensity, item 1b measures average pain intensity over the past 4 weeks, and item 1c assesses the worst pain intensity experienced in the last 4 weeks. Responses are recorded on an 11-point scale ranging from 0 (=no pain) to 10 (=worst imaginable pain). By calculating the mean of the three items multiplied by 10, values ranging from 0 to 100 are obtained. According to Von Korff et al. (1992; [66]), scores up to 49 indicate low pain intensity, while scores of 50 and above indicate high pain intensity. Since items 1b and 1c refer to the past month and thus cover a period beyond the timeframe of interest for the discharge assessment (i.e., the last two weeks), current pain intensity (item 1a) was considered separately in the calculations.

*Fibromyalgia Symptom Questionnaire*. The third item of the Fibromyalgia Symptom Questionnaire [20] (German version by [67]) was used to assess the number of painful body regions over the past seven days. Respondents can indicate pain in 19 different body regions or select the option “no pain in any of the listed body regions”. The selected body regions are then summed up for each individual.

*Multidimensional Fatigue Inventory (MFI-20)*. The Multidimensional Fatigue Inventory (MFI-20; [68]; German version by [69]) appears to be a suitable measurement tool for FMS according to Ericsson and Mannerkorpi (2007; [70]) and Ericsson et al. (2013; [71]). Fatigue is measured based on 20 items. The questionnaire assesses five dimensions of fatigue (each with four items) with the subscales general fatigue (example item: “I feel fit.”), mental fatigue (example item: “I can concentrate well.”), physical fatigue (example item: “Physically, I can do a lot.”), reduced motivation (example item: “I feel like doing all sorts of nice things.”), and reduced activity (example item: “I feel active.”). Responses are on a 5-point scale (1 = yes, that applies to 5 = no, that does not apply) and summed up to yield a subscale score ranging from 4 to 20. The total fatigue score is calculated by summing the values of the subscales (range 20–100). Higher (total) sum scores indicate higher levels of fatigue. Although Smets et al. (1995; [68]) did not originally propose calculating a total score across all 20 items (instead of subscale scores), the use of the total score (in addition to subscale scores) seems more appropriate for evaluating fatigue in the clinical setting (due to poor model fit; [69,72,73,74]). Therefore, the total score was calculated for the current study.

#### 2.3.2. Psychometric Outcomes and Quality of Life

*Hospital Anxiety and Depression Scale (HADS).* Anxiety and depression, which can precede or result from FMS, contribute to reduced QoL in FMS patients. The Hospital Anxiety and Depression Scale (HADS; [75]; German version by [76]), specifically developed for assessing anxiety and depression in patients with physical illnesses or (psychogenic) somatic symptoms in clinical settings, was utilized. This self-report questionnaire consists of two subscales, each with seven items, capturing the extent of anxious (e.g., “I feel panic-stricken.”) and depressive (e.g., “I feel happy.”) symptoms over the past week. Each item can score between 0 and 3 (e.g., 0 = not at all to 3 = most of the time). The sum scores of the HADS-A (Anxiety) and HADS-D (Depression) scales range from 0 to 21, with higher values indicating greater anxiety or depression. According to Zigmond and Snaith (1983; [75]), scores of 0–7 on each HADS subscale (HADS-A/D) are considered normal, 8–10 mild, and scores of 11 to 21 indicative of significant symptoms. While it is possible to calculate a total HADS score by summing the anxiety and depression items, an optimal cut-off value based on sensitivity and specificity calculations has not yet been determined [77]. Therefore, no overall score is calculated in the current study.

*Perceived Stress Scale-10 (PSS-10)*. Increased perceived stress, which is related to anxiety and depression, can reduce QoL—therefore, dealing with stress plays an important role with regard to the disease [78,79]. Hence, perceived stress was assessed using the Perceived Stress Scale-10 (PSS-10; [80]; German version by [81]), an internationally accepted instrument for measuring subjective stress levels. The ten questions relate to thoughts and feelings during the past month and have to be answered on a 5-point scale (1 = never to 5 = very often). The helplessness subscale consists of six items (example item: “How often have you felt that you were unable to control the important things in your life during the last month?”) and the self-efficacy subscale consists of four items (example item: “How often have you felt confident about your ability to handle your personal problems?”). Higher scores on the subscales and on the total score (sum of both subscales) indicate higher subjective stress levels. As the PSS-10 is not a diagnostic tool, there are no cut-off values.

*Quality of Life*. The Short-Form Health Survey 12 (SF-12; [82]; German version by [83]), a screening tool for assessing health-related QoL, comprises 12 items, assessing physical and mental aspects of health-related QoL over the past four weeks. The two subscale sum scores (containing 6 items each) represent the severity of physical and mental aspects of health-related QoL. The physical scale (example item: “I was limited in the kind of work or other activities.”) reflects physical functioning through the general perception of own health, physical and role functioning, and bodily pain. The mental scale measures mental QoL through role limitations due to emotional problems, general mental health, negative affect, and social functioning. Lower scores on the scales indicate poorer QoL, while higher scores indicate better QoL.

#### 2.3.3. Further Outcomes

*Work ability*. The subjective current work ability was assessed across all three measurement points using an 11-point scale (0 = complete inability to work, 10 = best imaginable work ability).

*Satisfaction with treatment*. At T2 satisfaction with the two-week inpatient treatment was assessed by using an 11-point scale (0 = not satisfied at all, 10 = very satisfied).

*Implementation in daily life*. At T3, it was also of interest whether specific methods and procedures learned during the clinic stay had been applied at home in the past six months. The item could be answered with yes or no, and in case of a negative response, a reason could be provided. Two open-ended questions were used to capture the obstacles and difficulties encountered in implementing the techniques in everyday life on one hand, and what would be helpful in applying the procedures in daily life on the other hand. Finally, patients were asked how often they still practiced several therapies (physical activity/movement therapy, mind–body stress reduction and relaxation techniques, implementation of dietary modifications, breathing exercises) in daily life after discharge (five-point scale: 0 = never, 4 = very often).

### 2.4. Procedure

All patients had to fill out the questionnaires for T1, T2, and T3. Filling out the T1 questionnaires took approximately 20–30 min, while the T2 and T3 questionnaires took about 15–25 min each. The T1 questionnaires, along with the patient information, were handed over to the patients on the day of admission or the following day. The T2 questionnaires were given to the patients on the day before discharge. After filling them out, patients returned them to the research team. The T3 questionnaires were sent to patients by mail six months later. If the T3 questionnaires were not returned to the research department within two weeks, patients were reminded by phone to fill them out and send them back. At times, questionnaire packages had to be resent because they had been lost either by the patients or in the mail.

### 2.5. Statistical Methods

Differences in the variables of interest between the three measurement time points were examined using one-factor repeated-measures analyses of variance (ANOVAs). In case of violation of the sphericity assumption, the Greenhouse-Geisser correction was applied. For significant main effects of measurement time point, pairwise post hoc comparisons were conducted using Bonferroni corrections. Since the statistical procedures required complete responses from each patient at all three measurement time points, there were deviations from the total sample size in the individual analyses of the variables. Furthermore, if patients had not fully answered a questionnaire for at least one measurement time point, resulting in the inability to calculate sum or subscale scores, they were also excluded from analyses. Data were analyzed using IBM Statistics (Version 30.0.0.0; Armonk, NY, USA [84])

## 3. Results

### 3.1. FMS Sample

The total FMS sample comprised N = 134 patients (127 female), aged between 22 and 80 years (M = 57.2, SD = 8.5). Sample characteristics can be found in Table 1. Completed T1 questionnaires were available for all included patients (N = 134), completed T2 questionnaires for 133 patients (with one missing), and completed T3 questionnaires were available for 101 individuals. Because the short- and long-term effects were of interest, the statistical analyses incorporated all three measurement time points (baseline [T1], discharge [T2], six-months follow-up [T3]). Therefore, for the final analysis, only patients were included for whom the T3 questionnaires were available, resulting in a total of 101 patients (97 female) with ages ranging between 40 and 78 years of age (M = 57.71, SD = 7.75). Reasons for the lack of T3 questionnaires include, that some patients could no longer be reached by phone after the T3 questionnaires had not been returned within two weeks (*n* = 12), some patients refused to complete the T3 questionnaires (*n* = 17), or they were unwell and therefore feeling unable to complete them (*n* = 4).

*Waterfiltered infrared-A WBH*. 20.5% (*n* = 17) of our patients did not receive hyperthermia, whereas 78.3% received either one (34.9%) or two (43.4%), and in 1.2% treatment had to be stopped. Due to the considerable difference in group size, a comparative statistical analysis was not possible.

### 3.2. Short- and Long-Term Changes in Main FMS Symptoms of Pain and Fatigue

*Pain*. For the analysis of overall pain intensity (items 1a, 1b, 1c of the Von Korff Grading of Severity questionnaire; [66]) *n* = 90 patients could be included. There was a significant main effect of measuring time point, *F*(2, 178) = 12.79, *p* < 0.001, η_p_^2^ = 0.13. Pairwise comparisons revealed a significant reduction in pain intensity from T1 (*M* = 71.52, *SD* = 12.44) to T2 (*M* = 63.04, *SD* = 14.69; *p* < 0.001, *d_z_* = 0.62) and from T1 to T3 (*M* = 64.43, *SD* = 18.01; *p* = 0.001, *d_z_* = 0.47). There was no change in overall pain intensity from T2 to T3 (*p* = 0.426, *d_z_* = 0.09). Looking at the current pain intensity (only item 1a; n = 94 patients), there was a significant main effect of measuring time point, *F*(2, 186) = 48.87, *p* < 0.001, η_p_^2^ = 0.34. Pairwise comparisons revealed a significant reduction in current pain intensity from T1 (*M* = 6.56, *SD* = 1.51) to T2 (*M* = 4.37, *SD* = 2.04; *p* < 0.001, *d_z_* = 1.23) and from T1 to T3 (*M* = 5.96, *SD* = 2.03; *p* = 0.013, *d_z_* = 0.34), and a significant increase from T2 to T3 (*M* = 5.96, *SD* = 2.03; *p* < 0.001, *d_z_* = 0.78). Changes in current and overall pain intensity can be depicted in Figure 1A,B.

Medical fasting was performed by approximately half of the participants (fasting: *n* = 44; no fasting: *n* = 46) and it was tested if there was a difference in overall pain intensity due to fasting. There was a significant interaction of measuring time point and if the participants fasted or not, *F*(2, 176) = 2.60, *p* = 0.050, η_p_^2^ = 0.03. Pairwise comparisons revealed for the fasting patients a significant reduction in overall pain intensity from T1 (*M* = 71.29, *SD* = 12.06) to T2 (*M* = 65.53, *SD* = 12.28; *p* = 0.008, *d_z_* = 0.47) and from T1 to T3 (*M* = 62.61, *SD* = 18.46; *p* = 0.006, *d_z_* = 0.57). There was no difference from T2 to T3 (*p* = 0.233, *d_z_* = 0.19). For the non-fasting patients there was a significant reduction in pain from T1 (*M* = 71.74, *SD* = 12.93) to T2 (*M* = 60.65, *SD* = 16.46; *p* < 0.001, *d_z_* = 0.75), but not from T1 to T3 (*M* = 66.16, *SD* = 17.61; *p* = 0.062, *d_z_* = 0.37). Importantly, there was a significant increase in overall pain intensity from T2 to T3 (*p* = 0.022, *d_z_* = 0.32), see Figure 1D.

For the calculation of the number of painful body sites (Item 3, FSQ; [67]), *n* = 99 patients were included. There was significant main effect of time point, *F*(2, 196) = 6.01, *p* = 0.003, η_p_^2^ = 0.06, i.e., the number significantly decreased from T1 (*M* = 11.88, *SD* = 3.80) to T2 (*M* = 10.55, *SD* = 4.65; *p* < 0.001, *d_z_* = 0.31), whereas there was no difference between T2 to T3 (*M* = 11.21, *SD* = 4.08; *p* = 0.115, *d_z_* = 0.15) and T1 to T3 (*p* = 0.084, *d_z_* = 0.17). Figure 1C illustrates the changes.

*Fatigue*. For the symptom of fatigue, the total score of the MFI-20 [69] was analyzed (*n* = 83). There was a significant main effect of the measurement time point, *F*(2, 164) = 12.42, *p* < 0.001, η_p_^2^ = 0.13, showing a significant decrease from T1 (*M* = 71.14, *SD* = 13.38) to T2 (*M* = 63.31, *SD* = 17.09; *p* < 0.001, *d_z_* = 0.51) and from T1 to T3 (*M* = 68.01, *SD* = 18.14; *p* = 0.049, *d_z_* = 0.20) and a significant increase from T2 to T3 (*p* = 0.004, *d_z_* = 0.27). The changes in fatigue are illustrated in Figure 2.

### 3.3. Short- and Long-Term Outcomes Regarding Psychometric Outcomes and Quality of Life

*Anxiety and Depression*. For the anxiety subscale (HADS-A; [76]), data of *n* = 93 patients could be included. There was a significant main effect of the measurement time point, *F*(2, 184) = 25.10, *p* < 0.001, η_p_^2^ = 0.21, i.e., anxiety was significantly reduced from T1 (*M* = 9.23, *SD* = 3.74) to T2 (*M* = 6.77, *SD* = 3.92; *p* < 0.001, *d_z_* = 0.64), increased from T2 to T3 (*M* = 7.73, *SD* = 4.03; *p* = 0.009, *d_z_* = 0.24), and was at T3 still significantly lower than at T1, *p* < 0.001 (*d_z_* = 0.39). These changes are shown in Figure 3A. For the depression subscale (HADS-D; *n* = 87), the main effect of the measurement time point was significant, *F*(2, 172) = 22.36, *p* < 0.001, η_p_^2^ = 0.21, showing that depression scores initially decreased significantly from T1 (*M* = 9.01, *SD* = 3.81) to T2 (*M* = 6.53, *SD* = 4.44; *p* < 0.001, *d_z_* = 0.60), increased significantly from T2 to T3 (*M* = 7.63, *SD* = 4.45, *p* = 0.006, *d_z_* = 0.25), but were still significantly lower at T3 compared to T1 (*p* < 0.001, *d_z_* = 0.33). Figure 3B shows the changes in the HADS-D.

*Perceived stress, helplessness, and self-efficacy*. The total perceived stress score (*n* = 93) of the PSS-10 [81] showed a significant main effect of the measurement time point, *F*(2, 184) = 7.26, *p* < 0.001, η_p_^2^ = 0.07, with no differences from T1 (*M* = 31.80, *SD* = 7.40) to T2 (*M* = 31.20, *SD* = 7.87; *p* = 0.187, *d_z_* = 0.08), but a significant reduction from T2 to T3 (*M* = 29.46, *SD* = 7.98, *p* = 0.013, *d_z_* = 0.22) and from T1 to T3 (*p* = 0.002, *d_z_* = 0.30). Changes across the three measurement time points are illustrated in Figure 4A.

For the *helplessness* subscale (*n* = 96), the main effect of the measurement time point was significant, *F*(2, 190) = 10.17, *p* < 0.001, η_p_^2^ = 0.10, i.e., helplessness did not change significantly from T1 (*M* = 19.61, *SD* = 4.90) to T2 (*M* = 19.41, *SD* = 4.99; *p* = 0.521, *d_z_* = 0.04), but was significantly reduced from T2 to T3 (*M* = 17.79, *SD* = 5.21; *p* < 0.001, *d_z_* = 0.32) and at T3 reduced compared to T1 (*p* < 0.001, *d_z_* = 0.36). Changes in the helplessness subscale are depicted in Figure 4B.

For the *self-efficacy* subscale (*n* = 95), there was a significant main effect of the measurement time point, *F*(2, 188) = 2.99, *p* = 0.050, η_p_^2^ = 0.03, showing that perceived self-efficacy significantly increased from T1 (*M* = 11.69, *SD* = 3.06) to T2 (*M* = 12.21, *SD* = 3.16; *p* = 0.017, *d_z_* = 0.17) and from T1 to T3 (*M* = 12.32, *SD* = 3.38; *p* = 0.039, *d_z_* = 0.20), and did not differ between T2 and T3 (*p* = 0.701, *d_z_* = 0.03). Changes are illustrated in Figure 4C.

*Health-related QoL*. For the physical scale score (*n* = 78) of the SF-12 [83] there was a significant main effect of the measurement time point, *F*(2, 154) = 12.99, *p* < 0.001, η_p_^2^ = 0.14, showing a significant improvement in physical QoL from T1 (*M* = 30.12, *SD* = 7.71) to T2 (*M* = 34.46, *SD* = 7.70; *p* < 0.001, *d_z_* = 0.57) and from T1 to T3 (*M* = 32.85, *SD* = 9.05; *p* = 0.006, *d_z_* = 0.33), and no changes between T2 and T3 (*p* = 0.063, *d_z_* = 0.19). Changes in the physical scale score are illustrated in Figure 5A. For the mental scale score (n = 78), there was a significant main effect of the measurement time point, *F*(2, 154) = 9.61, *p* < 0.001, η_p_^2^ = 0.11, with no difference between T1 (*M* = 33.14, *SD* = 13.23) and T2 (*M* = 33.76, *SD* = 12.87, *p* = 0.549, *d_z_* = 0.05), but a significant increase from T2 to T3 (*M* = 38.44, *SD* = 13.52; *p* = 0.001, *d_z_* = 0.35) and from T1 to T3 (*p* < 0.001, *d_z_* = 0.40). Changes are graphically represented in Figure 5B.

*Subjective current work ability*. When considering the subjective current work ability of *n* = 61 patients, a significant main effect of time point was observed, *F*(2, 120) = 5.51, *p* = 0.005, η_p_^2^ = 0.08 with significantly higher subjective work ability at T2 (*M* = 4.46, *SD* = 2.96) compared to T1 (*M* = 3.51, *SD* = 2.56; *p* = 0.002, *d_z_* = 0.34), and T3 (*M* = 4.21, *SD* = 2.69) compared to T1 (*p* = 0.022, *d_z_* = 0.27), but no change from T2 to T3 (*p* = 0.404, *d_z_* = 0.09). The changes are depicted in Figure 6.

All effect sizes are reported in the Appendix A.

*Satisfaction with treatment*. The average satisfaction with treatment at T2 was at 8.55 (*SD* = 1.97; *n* = 84).

*Implementation in daily life*. When asked whether individual methods and procedures learned in the clinic had been used at home in the past six months, 88.9% of the patients (*n* = 102) answered with “yes” and 11.1% answered with “no” (reasons given included: Lack of motivation, lack of help, very severe pain, acute rheumatic attack). Regarding the practicing of individual therapy methods, results show that most patients still practiced at least *sometimes* (up to *very often*): physical activity/movement therapy (84.3%), mind–body stress reduction and relaxation techniques (75.5%), implementation of dietary modifications (91.2%), and breathing exercises (77.5%).

The item or open question on obstacles and difficulties in implementing procedures in everyday life was answered by 61.8% of the n = 102 patients. These people most frequently mentioned the following obstacles and difficulties: Lack of time and time required for the procedures (34.9%), exhaustion and lack of energy due to work and everyday tasks (22.2%), lack of nearby offers/available appointments (11.1%), lack of motivation/lack of consistency and self-care (14.3%) and lack of utensils/implementation alone does not work/lack of group (11.1%). The item or open question on what would be helpful in applying the procedures in everyday life was answered by 53.9% of the *n* = 102 patients. The most common responses were: more time for themselves (32.7%) or fixed times/schedule (7.3%), professional guidance/contact person (20.0%) or a fixed group (5.5%).

## 4. Discussion

This observational study has three results we think to be important:

Firstly, our data show that a two-week integrative multimodal treatment program has a positive impact on patients who suffered from the chronic disease FMS for many years.

Secondly, significant improvements in core symptoms such as fatigue and pain were observed in the short and long term, with medical fasting identified as a possible factor in prolonging those positive effects.

Thirdly, after the inpatient stay, there was a reduction in psychological symptoms such as anxiety, depression, the subjective perception of stress and an improvement of self-efficacy. Combined with the reduction in fatigue and pain, this had a positive impact on our patients` overall quality of life.

Previous studies have already shown positive effects of individual complementary or integrative medical therapies for FMS (e.g., tai chi [37], yoga [38], meditation [59], nutritional therapy [60]), but the application of a comprehensive integrative medical concept in the treatment of FMS has not yet been scientifically investigated. The present study evaluated whether the core symptoms of pain and fatigue as well as the quality of life and further psychometric outcomes (e.g., anxiety, depression, and stress) of patients with FMS can be improved by a two-week inpatient multi-modal stress reduction and lifestyle modification program. Therapies are individually adapted to the patients and their disease characteristics (claimed by, e.g., [79,85]). Focus was at changes in the above-mentioned core symptoms and constructs both in the short-term (i.e., between admission and discharge), and in the long-term (i.e., between admission and follow-up or discharge and follow-up). Short-term improvements could be observed regarding overall pain intensity in the fasting and non-fasting group, current pain intensity, number of painful body sites, total fatigue score, anxiety, depression, self-efficacy, the physical scale of health-related QoL, and working ability. Long-term improvements were shown for overall pain intensity (especially in the fasting group), current pain intensity, fatigue, anxiety, depression, subjectively perceived stress, helplessness, the physical and mental scale of health-related QoL, and working ability. Reduction in the number of painful body sites and overall pain intensity in the non-fasting group improved only in the short term. Reduction in perceived stress, helplessness and mental QoL improved only in the long term (not in the short term).

The slightly differing results with regard to pain, i.e., current pain intensity increased significantly between T2 and T3, while overall pain intensity did not, could be due to the fact that different time periods were covered by the items: While the current pain intensity only recorded the pain at the time of completing the questionnaire, the overall pain intensity recorded both the current pain intensity and the average and greatest pain intensity during the last four weeks. However, both measures still show a significant improvement at T3 compared to T1. As the reference periods (last four weeks) in the questionnaires were constant across all measurement time points, the actual period of interest (at discharge the last two weeks) was not asked at every point in time. This poses a particular problem for the interpretation of the results at discharge, as not only the last two weeks in the clinic were included, but also the two pre-inpatient weeks. For this reason, with regard to overall pain intensity, the results of those analyses relating to the time of discharge should be given less consideration and instead the result of the comparison between admission—follow-up period should be weighted more heavily. In this way, the four weeks before the hospital stay are compared with the four weeks before the third measurement (approx. 5 months after the hospital stay). Looking at those results, there is a clear decrease in overall pain intensity as well as current pain intensity, arguing for long-term effects of our two-weeks inpatient stay (medium to large effect size). Looking closer at one of our individual treatments, results furthermore showed, that only the medical fasting patient group had not only short-, but long-term reduction effects in overall pain intensity. The non-fasting group, however, showed after a decrease in pain at discharge a significant increase in pain intensity at follow-up. Medical fasting can therefore be one aspect to help prolonging this effect. However, it is important to note that not all individuals may be suitable candidates for fasting, as there are potential contraindications that must be taken into consideration, such as eating disorders or diabetes type I. Therefore, the decision regarding the appropriateness of fasting should be made on a case-by-case basis by the treating physician. These results are in line with previous studies on therapeutic fasting in FMS patients also showing a positive effect on pain perception and furthermore on subsequent eating behavior [43,44,45]. Müller et al. (2001; [86]) could also find evidence for long-term improvements in pain through fasting (followed by a vegetarian diet). Fasting treatments may support motivation and self-efficacy in modifying their lifestyle into a health-promoting lifestyle [45,87]. Fasting implies the need for self-restraint in order to maintain it, which is inherently a challenge to self-management strategies [43,88] and probably one reason for the positive influence of fasting on quality of life in FMS patients. Previous research furthermore showed a mood-enhancing effect of caloric restriction and fasting, probably—among other things—due to an increased central serotonin availability that has been described experimentally (e.g., [89,90]). Through mood enhancement, fasting has the potential to alleviate core FMS symptoms such as pain [91]. Fasting effects could also be explained by antioxidant capacity released by fasting, as FMS patients seem to produce more damaging free radicals than healthy controls affecting their nervous system [92]. Medical fasting may lead to long-term reductions in pain intensity through a variety of interrelated physiological mechanisms. One well-documented effect is the reduction in systemic inflammation. Fasting has been shown to decrease levels of pro-inflammatory cytokines such as *IL-6*, *TNF-a*, and C-reactive protein (CRP), which play a central role in chronic pain pathophysiology [93,94,95]. Overall, it appears that a fasting intervention is often seen by patients as a starting point for habit change in general—dietary change in particular—probably leading to a more sustained reduction in FMS symptoms.

The current pain intensity could provide more reliable information regarding the change in pain across all three measurement points. However, it only represents a snapshot of the pain and therefore does not necessarily relate to our time period of interest (e.g., two-weeks inpatient stay). Despite those minor short-comings, results show a short-term and long-term reduction in current pain intensity at discharge (with a very large effect size). The item for the number of painful body sites covered the last seven days, making the results for all three measurement time points overall more reliable and comparable. There was a short-term reduction in this pain-related measurement (with small to medium effect size). Even if the number of painful body sites has increased again six months after discharge (and after resuming pre-inpatient everyday activities), a lasting improvement in pain intensity can still be observed after 6 months.

We can conclude, that pain could be significantly reduced by the treatment program, tends to return to the baseline level six months after the hospital stay, but is still comparatively better than at admission. Medical fasting can be one additional therapy to extend the duration of the effect.

Similar to the core symptom of pain, fatigue (total score) was significantly reduced from admission to discharge, but increased again from discharge to six months later (medium to large effect size). Importantly, it was still significantly reduced compared to the time of admission. There are many possible reasons for this renewed increase in symptoms after the significant improvements achieved during the hospital stay. On the one hand, the patients were again directly confronted with the stress and challenges of everyday life, which certainly led to an increase in symptoms; on the other hand, the simultaneous abrupt reduction in helpful treatments may have contributed to the fact that the successes achieved could not be fully maintained. It is obvious that the scope of therapies of inpatient treatment can be challenging to transfer to everyday life at home. Despite the increase in symptoms, the majority of patients stated that they had used individual methods and procedures learned in the clinic at home in the last six months, which indicates that at least some of the improvements achieved in the clinic could be maintained (as can be seen in the still reduced fatigue at follow-up compared to admission). The results of various previously mentioned studies on pain-relieving integrative medical procedures such as tai chi [37], Qigong [39], yoga [38] or meditation [59] emphasize that these procedures must be well practiced and practiced regularly in order to be able to relieve pain permanently. From qualitative interview studies we have conducted on the implementation of learned techniques in everyday life, it is known that it takes some time for the individuals to find the most helpful techniques, to develop a routine in practicing, and to set priorities in everyday life. Attending a day clinic program where patients can practice techniques over a longer period of time in a group setting under professional guidance (mind–body medicine in integrative and complementary medicine [MICOM]-day clinic Bamberg) seems to be very helpful in this respect [96,97].

For the FMS core symptoms of pain and fatigue, it can be summarized that they can be significantly reduced in the short term by the two-week integrative multimodal treatment concept, but that this treatment success could only partially be maintained six months after treatment. However, pain and fatigue were still better at follow-up compared to admission.

In addition, the subjectively assessed current ability to work, which was significantly increased by the therapy program, remained unchanged from discharge to follow-up, but was still significantly higher than at admission.

Regarding psychometric outcomes and quality of life, anxiety, depression, self-efficacy, and physical QoL (part of health-related QoL) improved significantly at discharge (large effect sizes). While anxiety and depression could be classified as “mild” on baseline according to the questionnaire, both were considered “unremarkable” on discharge. After the successful treatment, both deteriorated until follow-up, but were still significantly better compared to admission. Similar effects were shown for physical QoL. Physical QoL is measured by items on general health, physical pain, physical functioning, and physical role function. Combined with the result, that the number of painful body sites showed no difference between baseline and follow-up, this could be a reason why physical QoL could not be significantly improved overall (medium to large effect size). Expectation of self-efficacy was found to be one of the main factors influencing motivation and volition [98]. Therefore, the short- and long-term effects shown for self-efficacy (small effect sizes) are relevant indicators for long-term effects in all other investigated domains. Perceived stress, helplessness, and mental QoL (part of health-related QoL) remained unchanged from baseline to discharge. From discharge to follow-up, however, there was a significant improvement. To actually perceive a reduction in stress or in perceived helplessness it is necessary to change things for a longer period of time. Therefore, changes only become visible at follow-up. The improvements in perceived stress, helplessness, and mental QoL may have contributed to a reduction in long-term pain intensity or vice versa [78].

Anxiety and depression decreased together with pain and fatigue between baseline and discharge and deteriorated again together with pain and fatigue in the discharge-follow-up period. This can be seen as further support of the interaction relationship between chronic pain and depression [15] or pain perception and anxiety [99], although the direction of causality remains unclear.

Long-term effects reported six months after discharge compared to baseline could be attributed to the fact that helpful integrative methods continued to be used in the home setting. It is possible that the reduction in stress, anxiety, and depression reflects the effects of the psychosocial approaches (e.g., mindfulness and relaxation exercises) that were used as part of mind–body therapy during treatment and continued at home. Similarly, Müller et al. (2004; [28]) reported in their study on an outpatient multimodal group therapy concept for FMS, that symptoms were alleviated both in the short and long term, i.e., up to six months later, particularly at the psychological symptom level. Psychotherapeutic elements such as relaxation and body awareness exercises, resource work, and strengthening coping skills formed an important part of the therapy. Furthermore, there was a relevant change in self-efficacy between admission and follow-up, which is a basis for behavioral change and motivation [96]. Studies suggest that short-term positive effects of an (inpatient) intervention program can be attributed primarily to the intervention itself and/or externally motivated health behavior during the inpatient stay/intervention. In contrast, changes in health behavior and their maintenance after discharge/the end of the intervention depend on the patient’s intrinsic motivation and perception of self-efficacy [29,100,101,102]. In the patients studied here, it may be unrealistic to expect such a rapid increase in self-efficacy after a long period of suffering and unsuccessful therapies. Rather, even after the inpatient stay, more extrinsic motivational factors and time may still be necessary to develop intrinsic motivation to change and self-efficacy expectations, i.e., the perceived competence to initiate and maintain change [103]. However, our patients could significantly change their perception of self-efficacy and had higher perceived self-efficacy even at follow-up. Self-management, which plays an important role in FMS as a chronic illness [79,104], can be based on the Transtheoretical Model (TTM) of change [105]. This describes the stages of change as unintentionality, intention formation, preparation, action, and maintenance. In the preparation phase, for example, the patient realizes the importance of his/her own initiatives and perseverance for the success of the treatment. In the action phase, applications and procedures are carried out independently, but are not yet stabilized. Only in the maintenance phase behaviors exist that contribute to the regular use of helpful procedures even under (difficult) everyday conditions [104]. Applied to the current study and its results, this concept suggests that (some) patients have not yet reached the maintenance stage despite the successes achieved through the inpatient stress reduction and lifestyle modification program. Instead, they appear to remain in one of the earlier phases. More than half of the patients reported obstacles (such as exhaustion, lack of motivation, and lack of time) that made it difficult to apply the procedures they had learned in daily life. This may explain why the procedures were applied irregularly after discharge, as the two-week inpatient stay was too short for regular practice. The lack of continuous guidance and uncertainty regarding correct implementation meant that patients were probably unable to maintain the long-term benefits of the procedures. For this reason, it is recommended that patients seamlessly follow the inpatient program with a day clinic stay (MICOM-day clinic Bamberg). The day clinic creates a gradual transition from the clinic to everyday (working) life over a period of ten weeks. The focus is on practicing and consolidating of techniques and removing obstacles for the implementation of the learned contents. Patients are trained once a week for six hours in the implementation of a health-promoting lifestyle and given practical guidance, whereby naturopathic and complementary medical procedures are also used. It promotes patients’ sense of self-efficacy and contributes to the development of helpful routines through the direct connection to everyday life, the exchange with other affected people, and the professional guidance with the possibility of help and queries. Patients are thus motivated over a longer period of time to incorporate their own and newly discovered health resources into their everyday lives.

## 5. Strengths and Limitations

This was the first time to evaluate a two-week inpatient multi-modal integrative treatment concept for FMS patients with longstanding disease in Germany using a comprehensive set of validated questionnaires on relevant outcomes. The evaluation covered not only physical symptoms but also aspects regarding quality of life and mental health (psychometric outcomes). More than 75% of the included patients completed the questionnaires at all three timepoints (admission, discharge and follow-up at 6 months after discharge) which is a remarkable response rate and therefore allows an assessment whether the improvements are long lasting.

The primary limitation of this study is its design as health services research making it an observational study without a control group. This absence of a comparator limits the ability to draw valid causal inferences and makes it difficult to distinguish the observed effects of the study from non-specific influences. Treatment independent contextual effects, such as therapeutic and medical attention, treatment expectations, and the inpatient setting, might have influenced the outcomes. In order to further increase the evidence for causal relationships between the effect of the treatment and the observed changes in the investigated variables and to minimize bias due to confounding factors, future studies should use a randomized controlled design. In addition, the multimodal approach makes it difficult to analyze the effects of individual interventions, as the patients did not receive exactly the same therapies with the same frequency due to the definition of individual treatment priorities. With regard to the patients examined, it should be noted that in addition to the FMS diagnosis, some of the patients suffer from concomitant diseases that can also have an influence on their overall health-condition. Since the influence of other diseases on the assessment of outcomes cannot be completely ruled out, the interpretation of the results should take into account the potential impact of comorbidities. Conditions such as depressive disorders, irritable bowel syndrome, or sleep disorders may additionally affect mood, fatigue, pain, and sleep for instance. However, the holistic concept is particularly useful in this context, as it considers all areas of life that are relevant to health and so reveals its full potential.

Further research with a sufficient number of patients for subgroup analyses is needed.

## 6. Conclusions

The present study shows for the first time that a two-week inpatient multi-modal comprehensive lifestyle-modification and stress reduction program reduced the symptoms of pain and fatigue in FMS patients with longstanding disease. It also improved important aspects of quality of life through better self-management of their illness. Although their sustainability still needs to be investigated further, present effects were mostly long-term, with only few being only short-term. The results of this two-week program seem particularly promising with regard to pain, as such a rapid improvement would not necessarily be expected due to the degree of chronicity [28]. Patients were furthermore very satisfied with the treatment. Therefore, after validation through RCTs, the concept might be considered as a complementary therapy component in medical guidelines and thus in the care of FMS patients. Since some improvements seem to be rather short-lived and patients seem to have problems in integrating the contents of the therapy program into their daily lives, it might be helpful to add a day-clinic program subsequently (1 day weekly for 10 weeks). There, the focus is on practicing and consolidating techniques and removing obstacles for the implementation of the learned contents. Particularly in combination with a subsequent day clinic stay, the integrative approach could also help to reduce symptoms and healthcare costs in the long term, as patients can learn and maintain a health-promoting lifestyle in everyday life.

## Figures and Tables

**Figure 1 biomedicines-13-02144-f001:**
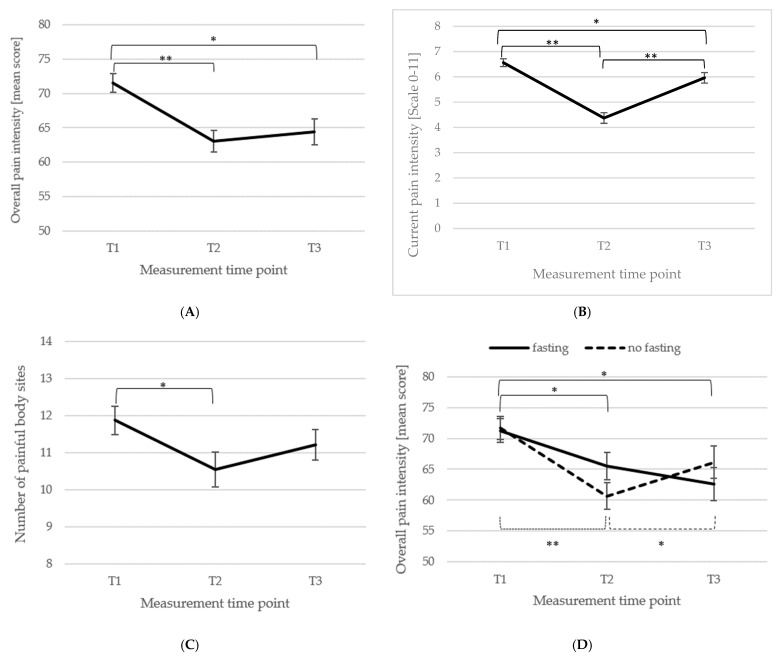
Mean overall pain intensity (**A**), current pain intensity (**B**), number of painful body sites (**C**); item 3 of the FMS; [67]) and overall pain intensity with relation to medical fasting are depicted (**D**) (±1 *SE*). *Note*: T1 = baseline, T2 = discharge, and T3 = follow-up; *n* (**A**) = 90, *n* (**B**) = 94, *n* (**C**) = 99, n (**D**) = 90; ** *p* < 0.001, * *p* < 0.05.

**Figure 2 biomedicines-13-02144-f002:**
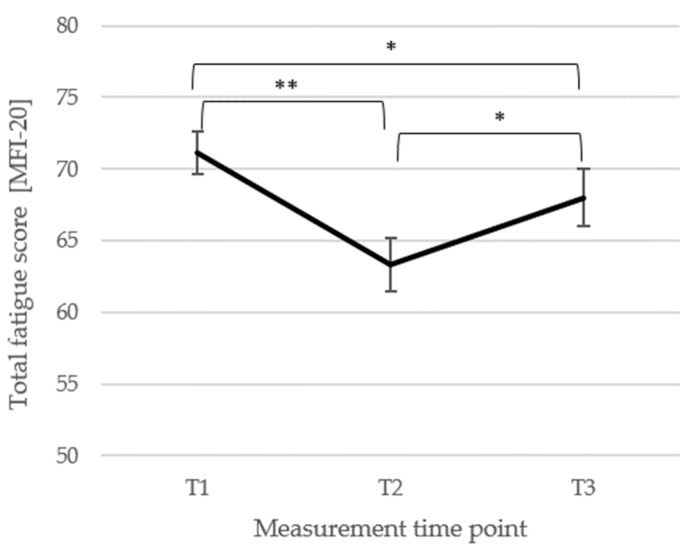
The total fatigue score (MFI-20; [69]) is depicted (±1 SE). *Note*: T1 = baseline, T2 = discharge, and T3 = follow-up; *n* = 83; ** *p* < 0.001, * *p* < 0.05.

**Figure 3 biomedicines-13-02144-f003:**
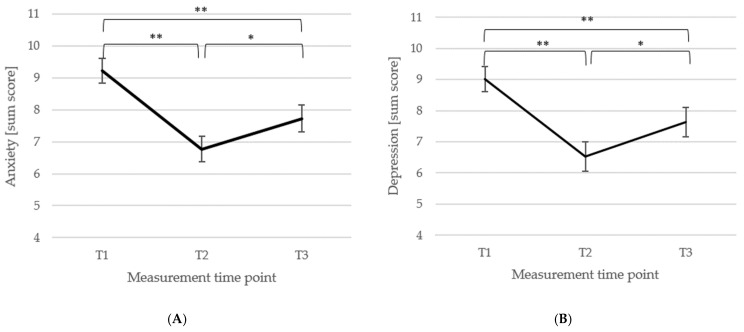
Means (±1 *SE*) for all three measurement points (T1, T2, T3) of the anxiety ((**A**); HADS-A) and depression ((**B**); HADS-D) sum score (HADS; [76]) are depicted. *Note*: T1 = baseline, T2 = discharge, and T3 = follow-up; *n* (**A**) = 93, *n* (**B**) = 87; ** *p* < 0.001, * *p* < 0.05.

**Figure 4 biomedicines-13-02144-f004:**
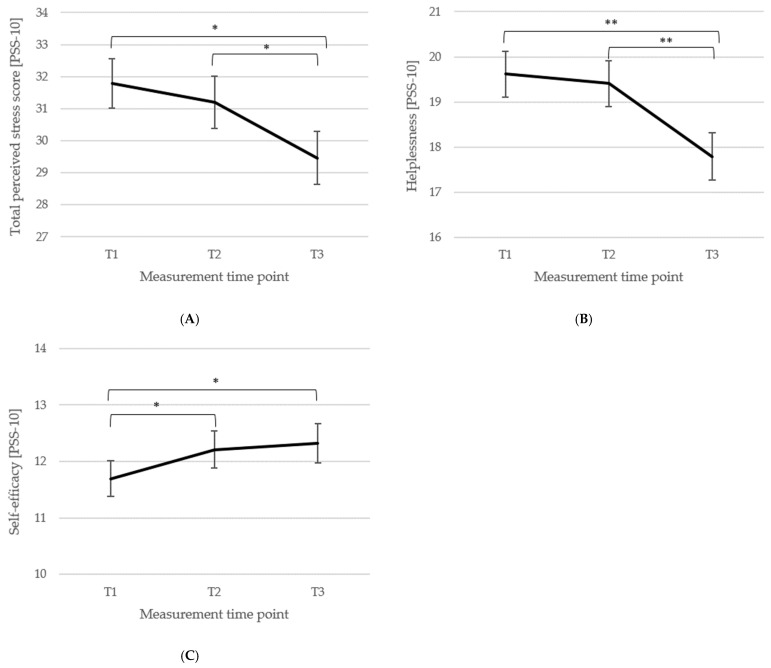
Means (±1 *SE*) for all three measurement points (T1, T2, T3) of total perceived stress (**A**), helplessness (subscale; (**B**)), and self-efficacy (subscale; (**C**)) of the PSS-10 [81] are depicted. *Note*: T1 = baseline, T2 = discharge, T3 = follow-up; *n* (**A**) = 93, *n* (**B**) = 96, *n* (**C**) = 95; ** *p* < 0.001, * *p* < 0.05.

**Figure 5 biomedicines-13-02144-f005:**
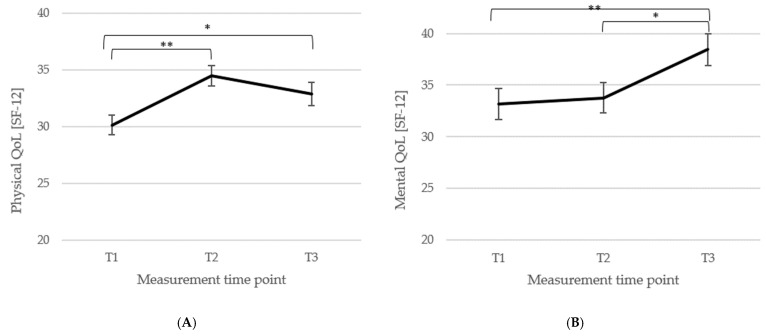
Means (±1 *SE*) for all three measurement points (T1, T2, T3) of physical (physical scale; (**A**)) and mental (mental scale; (**B**)) QoL (SF-12; [83]) are depicted. *Note*: T1 = baseline, T2 = discharge, T3 = follow-up; *n* (**A**) = 78, *n* (**B**) = 78; * *p* < 0.05, ** *p* < 0.001.

**Figure 6 biomedicines-13-02144-f006:**
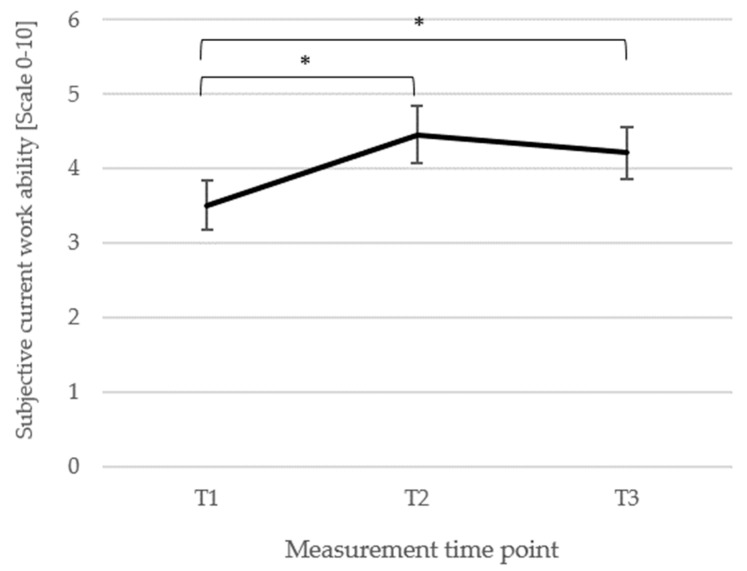
Means (±1 *SE*) for all three measurement points (T1, T2, T3) of the subjective current work ability are depicted. *Note*: T1 = baseline, T2 = discharge, and T3 = follow-up; * *p* < 0.05.

**Table 1 biomedicines-13-02144-t001:** Sample characteristics.

	Sample (*n* = 134)
**Sociodemographic characteristics**	
Sex	
female	127 (94.8%)
male	7 (5.2%)
Age (years)	57.2 ± 8.5
Educational level	
no school certificate	1 (0.7%)
secondary school	113 (84.3%)
A-level	17 (12.7%)
Professional qualification	
vocational training	78 (58.2%)
master training	32 (23.9%)
high school/university	10 (7.4%)
without/still in training	7 (5.2%)
Employment status	
full-time	13 (9.7%)
part-time/occasional work	35 (26.1%)
on sick leave	16 (11.9%)
unemployed	14 (10.5%)
retired (due to age)	22 (16.4%)
retired (due to health reasons)	32 (23.9%)
**Health related characteristics**	
Time since diagnosis (years)	9.2 ± 8.5
Body-Mass-Index (m/kg^2^)	29.3 ± 7.3
<19	0 (0%)
19–24	30 (22.7%)
24–29	53 (39.4%)
>30	51 (37.9%)

## Data Availability

Data will be provided by the corresponding author on request.

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
