# Peer review of "A Prospective Observational Study of a 2-Week Integrative Inpatient Therapy on Patients with Fibromyalgia Syndrome"

_biomedicines, 2025, doi:10.3390/biomedicines13092144_

Round 1
Reviewer 1 Report
Comments and Suggestions for Authors
Please read the attachment. Thank you.

Please read the attachment. Thank you.
Reviewer 2 Report
Comments and Suggestions for Authors
This study addresses a clinically relevant issue — the integrative and multimodal management of fibromyalgia, a complex and chronic condition. It is one of the few works to investigate a two-week program with multiple interventions and report outcomes up to six months later.
The topic is original and of interest to the scientific and clinical community, aligning with modern guidelines for managing functional somatic syndromes. However, substantial methodological limitations (especially the lack of a control group and heterogeneity of interventions) weaken the strength of the conclusions.
The absence of randomization and a comparator group (usual care or placebo) prevents any conclusions about causality.
Patients did not receive identical components of the program, making it unclear which elements contributed most to the observed effects.
All outcomes were self-reported and participants were aware of being in a study.
Some patients had associated conditions that could have influenced the outcomes.
Secondary analyses (e.g., fasting vs. non-fasting) have limited statistical power and borderline results.
Include a more detailed discussion of the impact of the lack of a control group and possible non-specific effects.
Clarify whether participants were instructed to maintain the learned practices after discharge, and how adherence was monitored.
Report effect sizes in the figures to aid clinical interpretation.
Discuss how comorbidities might have affected the results.
The English language throughout the manuscript is generally clear and understandable. However, some sentences in the discussion are overly long and could be made more concise for improved readability. Minor grammatical and stylistic editing is recommended to enhance clarity and flow. A professional language editing service could help polish the text further.
Reviewer 3 Report
Comments and Suggestions for Authors
Fibromyalgia is a debilitating pathology that interfere with daily activity with tremendous impact on quality of life for the patient. Etiology is unknown and treatment is usually a mix between pharmacological and non-pharmacological measures.
Proposed article is verry interesting due to the importance of the topic –multimodal therapy for longstanding fibromyalgia trying to cover multifaced nature of the disease (pain, sleep abnormality, mood disorder, etc.). Study is well structured, with explicit and easy to follow graphic representations. Results and discussions are clear, with references to other similar data. Putting together mind-body therapy with nutritional strategy, including fasting under medical supervision brings novelty to this study. Results are verry encouraging for both patients and medical stuff, since usually treating fibromyalgia patients is verry difficult due to low results.
I recommend acceptance of the manuscript after minor correction of the line 123 – different reference style.
Reviewer 4 Report
Comments and Suggestions for Authors
It is advised to change the meaning throughout the manuscript of the results including the abstract. The current format gives the expression of being a confirmative study regarding efficacy. But it is cannot be assumed especially in the situation that the authors did not evaluate with control individuals. What is the reason for not having controls? The authors can do a paired selection, or evaluate these individuals with different type of statistical analysis; but this will require specialized service, and rewritten of the manuscript.
NCT04927403 is also written in other manuscripts in the literature. https://www.sciencedirect.com/science/article/pii/S0022399923004245 I did not clear understand the reason for the same register in multiple studies with different pathologies. The authors should have a separated register in ther institution. The register of a clinical trial should be associated with the clinical protocol in the institution. Please provide the IRB (Ethics Committee of the Bavarian State Medical Association in Munich) of the institution too.
Regarding the statistics. ANOVA was used, but we do not have the report of the effect sizes, and also the authors should clarify. Please, write the software used for the statistics, edition, and city of development.
Please provide a figure with the clinical design of the study. It is still unclear what was the intervention. Try to do like a timeline for easy comprehension. Also, be specific of the types of therapy used.
The subgroup analysis on fasting is intriguing but underpowered. The interaction effect (p = .050) is borderline significant. This should be interpreted cautiously and framed as exploratory. Were there baseline differences between fasting and non-fasting groups?
The title of the manuscript should be revised to better reflect the nature of the study. The current use of the word “effects” implies a causal relationship, which is not appropriate for an observational study without a control group. Additionally, the term “integrative multimodal inpatient therapy” is overly broad and should be clearly defined early in the manuscript. While the study later outlines the components of the intervention—such as medical fasting, mind-body medicine, hydrotherapy, and lifestyle counseling—this should be briefly summarized in the abstract and introduction to ensure clarity for readers unfamiliar with the specific therapeutic model used.
To enhance transparency and adherence to reporting standards, a CONSORT-style flow diagram should be included as supplementary, non-publishable material. This diagram would visually summarize the patient recruitment process, including the number of individuals assessed for eligibility, enrolled, lost to follow-up, and included in the final analysis. Providing this visual aid would help readers quickly understand the study’s participant flow and improve the overall clarity of the methodology.
The abstract would benefit from the inclusion of specific numerical results to enhance clarity and impact. Currently, it provides general statements about improvements in symptoms such as pain and fatigue, but lacks quantifiable data. Including key statistics—such as the percentage reduction in pain intensity or fatigue scores from baseline to follow-up—would allow readers to better understand the magnitude of the observed changes. For example, stating that “pain intensity decreased by X% and fatigue by Y% at six-month follow-up” would make the findings more concrete and compelling.
Round 2
Reviewer 2 Report
Comments and Suggestions for Authors
Thank you for the revised version of your manuscript. I appreciate the efforts made to improve the discussion and address several of the previous comments. The topic remains highly relevant, and your observational study provides a valuable perspective on integrative, multimodal treatment approaches for fibromyalgia. That said, while some improvements are evident, important limitations remain either insufficiently addressed or still affect the strength of the conclusions.
The expansion of the discussion regarding the lack of a control group is welcome, and the revised text does acknowledge the limitations in attributing causality. However, the issue of non-specific effects, such as therapeutic attention, expectancy, and placebo-like contextual influence, is still not discussed in sufficient depth. Given the fully open-label nature of the intervention and the self-reported outcomes, a more transparent acknowledgment of these factors is essential. I encourage the authors to further strengthen this point to avoid overinterpreting associations as evidence of efficacy.
Regarding the maintenance of therapeutic practices post-discharge, the revised manuscript does now include a brief mention of patient-reported adherence data from the follow-up questionnaire. This addition improves the interpretation of long-term outcomes. However, the adherence results are not quantified or clearly presented. Even a brief table or percentage breakdown (e.g., number of participants reporting continued use of relaxation, physical activity, etc.) would add valuable context.
The inclusion of effect sizes in the discussion section is appropriate, and although not presented in the figures or legends, the updated narrative does refer to the magnitude of the effects. This partially addresses the concern, though including a supplemental table listing effect sizes for each key outcome variable would improve clarity and support clinical interpretation.
The brief addition related to comorbidities (“favorable outcomes are also anticipated in cases of comorbidities”) remains speculative and is not supported by specific analysis or subgroup stratification. While subgroup analysis may not be feasible due to sample size, a more nuanced commentary on how comorbidities might confound interpretation—especially regarding fatigue, mood, or sleep—would help balance the discussion.
The tone of the conclusion has been moderated slightly, and this revision is appreciated. Still, some statements remain too strong for an observational study without a comparator group (e.g., the suggestion that integrative therapy “should be a permanent component of routine care”). I would recommend replacing such phrasing with more cautious language (e.g., “may be a promising component” or “warrants further evaluation”).
Comments on the Quality of English LanguageThe English language is generally clear and understandable throughout the manuscript. However, some sentences, particularly in the discussion and conclusion, are overly long or use imprecise constructions that may hinder clarity. A light language revision focused on improving sentence structure and flow is recommended to enhance readability and ensure the arguments are communicated with precision.
Reviewer 4 Report
Comments and Suggestions for Authors
Satisfactory
Round 3
Reviewer 2 Report
Comments and Suggestions for Authors
The authors have addressed the reviewer’s comments appropriately across the previous rounds of revision. Although a few minor issues could still be refined, they do not compromise the integrity or scientific value of the manuscript. I support acceptance in its current form.